# Implicit Acceleration of Gradient Flow in Overparameterized Linear Models

### Abstract

We study the implicit acceleration of gradient flow in over-parameterized two-layer linear models. We show that implicit acceleration emerges from a conservation law that constrains the dynamics to follow certain trajectories. More precisely, gradient flow preserves the difference of the Gramian matrices of the input and output weights and we show that the amount of acceleration depends on both the magnitude of that difference (which is fixed at initialization) and the spectrum of the data. In addition, and generalizing prior work, we prove our results without assuming small, balanced or spectral initialization for the weights, and establish interesting connections between the matrix factorization problem and Riccati type differential equations.

## 1 Introduction

Understanding *over-parameterization* in deep learning is a puzzling question. Contrary to the common belief that over-parameterization may hurt generalization and optimization, recent work suggests that over-parameterization may actually bias the optimization algorithm towards solutions that generalize well, a phenomenon known as *implicit regularization* or *implicit bias*, and even accelerate convergence, a phenomenon known as *implicit acceleration*.

Recent work on the *implicit bias* in the over-parameterized regime (e.g. Gunasekar et al. (2018a;b); Chizat & Bach (2020); Ji & Telgarsky (2019b)) shows that gradient descent on unregularized problems finds *minimum norm solutions*. For instance, Soudry et al. (2018); Ji & Telgarsky (2019a) analyze linear networks trained for binary classification on linearly separable data and show that the predictor converges to a max-margin solution. Similar ideas have been developed for matrix factorization, yielding solutions with minimum nuclear norm (Gunasekar et al., 2017; Li et al., 2018) or low-rank (Arora et al., 2019a). It has also been shown that optimization methods which introduce multiplicative stochastic noise, such as dropout and dropblock, induce nuclear norm regularization (Cavazza et al., 2018) and spectral $k$-support norm regularization (Pal et al., 2020), respectively.

Recent work on the *implicit acceleration* of gradient descent for matrix factorization and deep linear networks (Arora et al., 2018) shows that when the initialization is sufficiently *small* and *balanced* (see Definition 2), over-parameterization acts as a pre-conditioning of the gradient that can be interpreted as a combination of momentum and an adaptive learning rate. They claim that acceleration for $\ell_p$-regression is possible only if $p > 2$, though there is no theory supporting such claim. Saxe et al. (2014) focused on $\ell_2$-regression with *balanced spectral initializations* (see Definition 3) and similarly concluded that depth may actually slow down the convergence. For two-layer linear networks, Saxe et al. (2019); Gidel et al. (2019) analyzed the dynamics of gradient flow and obtained explicit solutions under the assumption of vanishing spectral initialization, highlighting the sequential learning of the hierarchical components as a phenomenon that could improve generalization. Several recent papers (e.g Arora et al. (2019b); Du & Hu (2019); Du et al. (2018b)) have also analyzed the convergence behaviour of gradient descent in the over-parameterized setting, particularly for very wide networks and have concluded linear convergence when the initialization is gaussian or balanced. While a precise study of the connections between gradient descent and gradient flow dynamics for non-convex problems remains elusive, recent work (Franca et al., 2020) shows that discrete-time convergence rates can be derived from their continuous-time

Table 1: Relationships between our work and the state of the art.

|  | Small and Balanced | Imbalanced | Spectral | Non spectral |
|---|---|---|---|---|
| Gradient flow | Our work
Saxe et al. (2014)
Saxe et al. (2019) | Our work | Our work
Saxe et al. (2014)
Saxe et al. (2019) | Our work |
| Gradient descent | Arora et al. (2018)
Gidel et al. (2019) | None | Gidel et al. (2019) | Arora et al. (2018) |

counterparts via symplectic integrators. Therefore, our work focuses on the analysis of gradient flow as a stepping stone for future analysis of gradient descent.

In this paper, we present a new analysis of the implicit acceleration of gradient flow for over-parametrized two-layer neural networks that applies not only in the case of small, balanced, or spectral initialization but also extends to imbalanced and non-spectral initializations. We show that the key reason for the implicit acceleration of gradient flow is the existence of a *conservation law* that constrains the dynamics to follow a *particular path*.[1] More precisely, the quantity that is preserved by gradient flow is the difference of the Gramians of the input and output weight matrices, which in turn implies that the difference of the square of the norm of the weight matrices is preserved. The particular case where this difference is zero corresponds to the case of *balanced weights*, but the more general case of *imbalanced weights* also emerges as a conserved quantity and plays an important role. In particular, we show that acceleration can occur even in the case of $\ell_2$-regression as a result of imbalanced initialization. The reason this phenomenon was not previously observed in (Saxe et al., 2014; 2019; Gidel et al., 2019) is precisely due to the assumption of balanced initialization, which follows as a particular case of our analysis. Our work also establishes interesting connections with Riccati type differential equations. Indeed, some of our results have a similar flavor to those in (Fukumizu, 1998), while others are more general and provide an explicit characterization of the continuous-time convergence rate. In short, our work makes the following contributions.

1. In Section 2, we analyze the implicit acceleration properties of gradient flow for *symmetric* matrix factorization, providing a closed form solution and a convergence rate that depends on the eigenvalues of the data without the assumptions of spectral and small initialization.

2. In Section 3, we analyze the implicit acceleration properties of gradient flow for *asymmetric* matrix factorization with spectral initialization. We show that implicit acceleration emerges as a consequence of conservation laws that only appear in over-parameterized settings due to an underlying rotational symmetry.

3. In Section 4, we analyze the implicit acceleration properties of gradient flow for asymmetric matrix factorization with an arbitrary initialization. We make connections with Riccati differential equations, obtaining a more general characterization of the convergence rate and establish an interesting link with explicit regularization.

## 2 Gradient Flow Dynamics for Symmetric Matrix Factorization

In this section, we analyze and compare the dynamics of gradient flow,[2]
$$\dot{X}(t) = -\nabla_X \ell(X(t)), \tag{1}$$
when applied to two problems. The first one is learning a symmetric one-layer linear model
$$\min_{X \in \mathbb{R}^{m \times m}} \left\{ \ell(X) \equiv \tfrac{1}{2}||Y - X||_F^2 \right\}, \tag{2}$$
where $Y \in \mathbb{R}^{m \times m}$ is a given data matrix that one wishes to approximate by $X \in \mathbb{R}^{m \times m}$. The second one is learning its over-parameterized symmetric matrix factorization counterpart
$$\min_{U \in \mathbb{R}^{m \times k}} \left\{ \ell(U) \equiv \tfrac{1}{2}||Y - UU^T||_F^2 \right\}. \tag{3}$$

---

[1]A quantity $\mathcal{Q}(x(t))$ is said to be conserved by the flow $\dot{x}(t) = f(x(t))$ if it remains constant through dynamical evolution, i.e., $\frac{d}{dt}\mathcal{Q}(x(t)) = 0$. For example, in mechanics the sum of potential and kinetic energies remains constant for a conservative system. A conservation law is usually a consequence of an underlying symmetry (Noether's theorem). In optimization, this can be seen as a constraint $\mathcal{Q}(x) = \mathcal{Q}_0$ that is automatically satisfied without having to explicitly enforce it.

[2]Gradient descent, $x_{n+1} = x_n - \eta\nabla\ell(x_n)$, is simply an explicit Euler discretization of (1).

We show that the dynamics of the linear model converge at a rate $O(e^{-t})$, while the over-parameterized model has a rate of $O(e^{-4t|\sigma_i|})$, where $\sigma_i$ is the $i$th eigenvalue of the data matrix $Y$. Therefore, different spectral components are learned at different rates, which can be faster or slower than the non-overparameterized model depending on the eigenvalues of $Y$.

**Linear model.** Let us start with the trivial problem of learning a non-overparameterized linear model (2).[3] Applying the gradient flow (1) to problem (2) yields $\dot{X}(t) + X(t) = Y$ with $X(0) = X_0$. This is a linear differential equation whose unique solution is given by

$$X(t) = Y + (X_0 - Y)e^{-t}. \tag{4}$$

Thus, $\|X(t) - Y\|_F = e^{-t}\|X_0 - Y\|_F$ and $\lim_{t\to\infty} X(t) = Y$ at an exponential rate of $O(e^{-t})$.

For completeness, it will be interesting to consider the particular case in which $X$ is constrained to be positive semidefinite (PSD), i.e. $X \succeq 0$. In this case, notice that if $X_0 \succeq 0$ and $Y \succeq 0$, then $X(t) \succeq 0$ for all $t > 0$, hence the same dynamics and convergence rate still apply without having to enforce the PSD constraint. Otherwise, if $Y$ is not PSD, gradient flow is not directly applicable.

**Symmetric matrix factorization model.** Consider the more interesting case of learning a two-layer linear model with tied weights $U \in \mathbb{R}^{m \times k}$ formulated as the symmetric matrix factorization problem in (3). In classical low-rank matrix factorization, one assumes $k < m$. Here, we consider an over-parameterized formulation where $k > m$ plays the role of the number of hidden units (width). The gradient flow (1) on problem (3), for $U$, now yields $\dot{U} = 2(Y - UU^T)U$ with $U(0) \equiv U_0$. Letting $X(t) \equiv U(t)U(t)^T \succeq 0$ and $X(0) = U_0 U_0^T \succeq 0$ one can easily verify that

$$\dot{X} = \dot{U}U^T + U\dot{U}^T = 2(Y - X)X + 2X(Y - X) = 2YX + 2XY - 4X^2. \tag{5}$$

Equation (5) is known to be rank preserving, i.e. if $\text{rank}(X_0) = r \leq m$ and $X^* = \lim_{t\to\infty} X(t)$ exists then $\text{rank}(X^*) \leq r$. It is thus impossible to recover a solution of rank higher than the rank of the initialization.

We also note that (5) is a (matrix) differential equation of the *Riccati* type. Such equations often characterize dynamical systems behind least squares problems and have been extensively studied in the context of optimal control. Using results from this literature, we obtain (see Appendix A for the proof):

**Proposition 1.** *For any $X_0 \in \mathbb{R}^{m \times m}$, the solution to (5) exists and is given by*

$$X(t) = e^{2tY} X_0 \left( I + Y^{-1}(e^{4tY} - I)X_0 \right)^{-1} e^{2tY}, \tag{6}$$

*provided that $Y$ and the matrix inside the parenthesis above are invertible.*

This solution is derived for any $X_0$, while the over-parameterized model requires $X_0 = U_0 U_0^T \succeq 0$. Thus in using (6) as an analysis tool, it is important to keep in mind the set of allowable initializations. In what follows, we consider the spectral initialization (Saxe et al., 2019; Gidel et al., 2019), and show that the eigenspace of the data is preserved throughout the entire evolution of the learning dynamics.

**Definition 1 (Symmetric Spectral initialization).** *Let $Y = \Phi\Sigma\Phi^T$ be the eigendecomposition of the data. A spectral initialization is defined as $U_0 \equiv \Phi\Sigma_0^{1/2}$ and $X_0 \equiv U_0 U_0^T = \Phi\Sigma_0\Phi^T$ where $\Sigma_0 \succeq 0$ is a diagonal matrix.*

From the explicit solution (6), we can readily obtain a *convergence rate* for a spectral initialization (see Appendix B for a short proof).

**Corollary 1.** *If $Y = \Phi\Sigma\Phi^T = \sum_{i=1}^m \sigma_i \phi_i \phi_i^T$ is invertible and $X_0 = \Phi\Sigma_0\Phi^T = \sum_{i=1}^m \sigma_{0,i} \phi_i \phi_i^T$ is a spectral initialization, the solution to (5) is $X(t) = \Phi\Sigma(t)\Phi^T = \sum_{i=1}^m \sigma_i(t)\phi_i \phi_i^T$, where*

$$\sigma_i(t) = \frac{\sigma_{0,i}\sigma_i e^{4t\sigma_i}}{\sigma_i + \sigma_{0,i}(e^{4t\sigma_i} - 1)} = \sigma_i + \frac{\sigma_i(\sigma_{0,i} - \sigma_i)}{\sigma_i + \sigma_{0,i}(e^{4t\sigma_i} - 1)}, \tag{7}$$

---

[3]In a linear neural network, $Y$ plays the role of the input-output data correlation matrix and $X$ plays the role of the model's input-output map. In this trivial model, the input correlation matrix is assumed to be the identity as is the case when the data is whitened.

*provided the denominator is nonzero. Moreover, if $\tilde{Y} = \sum_{i=1}^{m} \max(\sigma_i, 0)\phi_i\phi_i^T = \Phi\tilde{\Sigma}\Phi^T$ is the projection of $Y$ onto the PSD cone, then for all initializations $X_0 \succeq 0$ such that $rank(\Sigma_0\tilde{\Sigma}) = rank(\tilde{\Sigma})$, $X(t)$ converges to $\tilde{Y}$, at a rate $O(e^{-4t\sigma_{\min}(Y)})$, where $\sigma_{\min}(Y) = \min_i |\sigma_i|$ is the smallest eigenvalue of $Y$ in magnitude.*

Note from (7) that the $i$th eigencomponent of $X$ converges at a rate of $O(e^{-4t|\sigma_i|})$, so that components with $4|\sigma_i| > 1$ are accelerated compared to the non-overparameterized case, components with $4|\sigma_i| < 1$ are slowed down, and if $4\sigma_{\min}(Y) > 1$ all components are accelerated. This result about different components of the network being learned at different rates by gradient flow is related in spirit to the result in (Saxe et al., 2019; Gidel et al., 2019) about sequential learning in the asymmetric case with spectral balanced initialization. Here the balancedness is enforced by construction.

Next, we derive the same convergence rate with a more general (non-spectral) initialization. The proof is in Appendix C and makes use of several interesting relations for Riccati differential equations.

**Proposition 2 (Convergence rate).** *Consider the eigenvalue decomposition $Y = \sum_{i=1}^{m} \sigma_i\phi_i\phi_i^T$. Let $\tilde{Y} = \sum_{i=1}^{m} \max(\sigma_i, 0)\phi_i\phi_i^T$ be the projection of $Y$ onto the PSD cone and $\hat{Y} = \sum_{i=1}^{m} |\sigma_i|\phi_i\phi_i^T$. For any initialization $X_0 \succeq 0$, assume that $I + \hat{Y}^{-1}(X_0 - \tilde{Y})$ and $Y$ are nonsingular. Then the solution $X(t)$ of (5) converges exponentially to $\tilde{Y}$ at a rate*

$$\left\|X(t) - \tilde{Y}\right\|_F \leq Ce^{-4t\sigma_{min}(Y)}, \tag{8}$$

*where $\sigma_{min}(Y)$ is the smallest eigenvalue of $Y$ in absolute value and $C > 0$ is a constant.*

It follows from Proposition 2 that the implicit acceleration for symmetric matrix factorization with spectral initialization can be extended to any positive semidefinite initialization $X_0$ provided $I + \hat{Y}^{-1}(X_0 - \tilde{Y})$ is invertible, which is an extension of the previous assumption on $X_0$, namely that $rank(\Sigma_0\tilde{\Sigma}) = rank(\tilde{\Sigma})$. The primary difference is that in the spectral initialization case we can derive the convergence rate for each eigenvalue of $X(t)$, while in general we can only obtain a global convergence rate of the solution.

## 3 Gradient Flow Dynamics for Asymmetric Matrix Factorization with Spectral Initialization

In this section, we analyze the dynamics of gradient flow for the more general asymmetric matrix factorization problem. We show that the implicit acceleration phenomenon is still present and provide an explanation for it based on a conservation law for the difference of the Gramians of the factors. We transform the dynamics to a canonical form and show that the solutions under the spectral initialization are diagonal and can be computed in closed form. The closed form solution reveals a convergence rate of $O(e^{-t\sqrt{4\sigma_i^2 + \lambda_{0,i}^2}})$, where $\sigma_i$ is the $i$th singular value of $Y$ and $\lambda_{0,i}$ defines the level of imbalance in the initialization for the $i$th component. As in the symmetric case, data matrices with large singular values induce implicit acceleration. However, in the asymmetric formulation, additional acceleration can be gained by choosing an imbalanced initialization.

**Asymmetric matrix factorization model.** Consider the asymmetric factorization problem:

$$\min_{U,V} \left\{\ell(U, V) \equiv \tfrac{1}{2}||Y - UV^T||_F^2\right\} \tag{9}$$

where $U \in \mathbb{R}^{m \times k}$, $V \in \mathbb{R}^{n \times k}$ and $k \geq n \geq m$. The gradient flow for this problem takes the form

$$\dot{U} = -\nabla_U \ell = (Y - UV^T)V, \qquad \dot{V} = -\nabla_V \ell = (Y - UV^T)^T U. \tag{10}$$

In what follows, We will make use of a conservation law for the difference of the Gramian matrices $U^TU$ and $V^TV$ which has been previously identified in (Du et al., 2018a; Arora et al., 2018). Previous works have used this conservation law to ensure balancedness under vanishingly small initialization. In contrast, our analysis is the first to highlight the role of imbalance and the resulting acceleration of gradient flow.

**Conservation law.** A straightforward calculation shows that (10) admits an invariant:

$$\mathcal{Q} \equiv U^T U - V^T V, \quad \frac{d\mathcal{Q}}{dt} = \dot{U}^T U + U^T \dot{U} - \dot{V}^T V - V^T \dot{V} = 0 \quad \Longrightarrow \quad \mathcal{Q}(t) = \mathcal{Q}(0). \quad (11)$$

The origin behind this conserved quantity $\mathcal{Q}$ is a global rotational symmetry of the system in (10), i.e., the system is invariant under the orthogonal group $O(k)$. To see this, consider the singular value decomposition $Y = \Phi \Sigma \Psi^T$ and, following (Saxe et al., 2019), define matrices $\bar{U}$ and $\bar{V}$ through

$$U = \Phi \bar{U} G^T, \qquad V = \Psi \bar{V} G^T, \quad (12)$$

where $G$ is an arbitrary element of $O(k)$. These transformed variables obey

$$\dot{\bar{U}} = (\Sigma - \bar{U}\bar{V}^T)\bar{V}, \qquad \dot{\bar{V}} = (\Sigma - \bar{U}\bar{V}^T)^T \bar{U}. \quad (13)$$

Note that these equations have *exactly the same form* as (10), up to a gauge freedom on the choice of $G$. Since $\mathcal{Q}$ is real and symmetric, it is diagonalizable by an orthogonal matrix. Therefore, we can choose $G$ to be the matrix that diagonalizes $\mathcal{Q}$—this is a gauge choice. Hence, from (11) we have

$$\bar{U}^T \bar{U} - \bar{V}^T \bar{V} = G^T \mathcal{Q}(t) G = \Lambda_{\mathcal{Q}} = \Lambda_{\mathcal{Q}_0} = \bar{U}_0^T \bar{U}_0 - \bar{V}_0^T \bar{V}_0, \quad (14)$$

where $\Lambda_{\mathcal{Q}_0}$ is the (constant) *diagonal matrix* containing the $k$ eigenvalues of $\mathcal{Q}(0) \equiv \mathcal{Q}_0$ (or $\mathcal{Q}$) which is completely specified by the initial conditions $U_0$ and $V_0$ alone. Note that the number of conserved quantities in $\Lambda_{\mathcal{Q}_0}$ depends on $k$, which is equal to the degree of over-parameterization. Though we do not assume balanced initialization in this paper, for further reference and comparison with prior work (Arora et al., 2018; Saxe et al., 2014; 2019) let us state its precise meaning since it relates to the conservation law.

**Definition 2 (Balanced initialization).** $(U_0, V_0)$ *is said to be balanced if* $\|\mathcal{Q}\|_F = \|\mathcal{Q}_0\|_F \leq \epsilon$ *for sufficiently small* $\epsilon > 0$, *i.e. the conserved quantity in* (11), *or equivalently* (14), *almost vanishes.*

Under the above transformation, the matrix problem with spectral initialization can be reduced to solving $k$ one-dimensional systems (one for each component). Proposition 3 provides a closed form solution and explicitly characterizes the evolution of each component along with its implicit acceleration (See Appendix D for the proof).

**Definition 3 (Asymmetric Spectral initialization).** *Let* $Y = \Phi \Sigma \Psi^T$ *be the SVD of the data. The spectral initialization is defined as* $U_0 = \Phi \bar{U}_0 G$, $V_0 = \Psi \bar{V}_0 G$, *and* $X_0 = U_0 V_0^T$, *where* $\bar{U}_0$ *and* $\bar{V}_0$ *are rectangular diagonal matrices and* $G$ *is any orthogonal matrix.*

**Proposition 3 (Implicit acceleration of asymmetric factorization with spectral initialization).** *Let* $Y = \Phi \Sigma \Psi^T = \sum_{i=1}^m \sigma_i \phi_i \psi_i^T$ *be the SVD of the data. The solution to* (10) *with spectral initialization* $X_0 = \Phi \Sigma_0 \Psi^T = \sum_{i=1}^m \sigma_{0,i} \phi_i \psi_i^T$ *yields* $X(t) = U(t)V(t)^T = \Phi \Sigma(t) \Psi^T = \sum_{i=1}^m \sigma_i(t) \phi_i \psi_i^T$, *where*

$$\sigma_i(t) = \frac{\sigma_i e^{2t\sqrt{4\sigma_i^2 + \lambda_{0,i}^2}} - 2C_i \lambda_{0,i}^2 e^{t\sqrt{4\sigma_i^2 + \lambda_{0,i}^2}} - 4\sigma_i \lambda_{0,i}^2 C_i^2}{e^{2t\sqrt{4\sigma_i^2 + \lambda_{0,i}^2}} + 8\sigma_i C_i e^{t\sqrt{4\sigma_i^2 + \lambda_{0,i}^2}} - 4\lambda_{0,i}^2 C_i^2}, \quad (15)$$

$\lambda_0 = \text{diag}(\bar{U}_0^T \bar{U}_0 - \bar{V}_0^T \bar{V}_0)$ *and* $C_i = C_i(\sigma_i, \lambda_{0,i}, \sigma_{0,i})$ *is a constant. Moreover, the* $i$th *eigencomponent of* $X(t)$ *converges to the* $i$th *eigencomponent of* $Y$ *at a rate* $O\big(e^{-t\sqrt{4\sigma_i^2 + \lambda_{0,i}^2}}\big)$.

**Implicit acceleration.** Recall from (4) that the convergence rate for the non-overparameterized problem in (2) is $O(e^{-t})$, which does not depend on the data or the initialization. It follows from (15) that the asymptotic behavior of the singular values of the over-parameterized solution is:

$$|\sigma_i(t) - \sigma_i| \simeq 2C_i(4\sigma_i^2 + \lambda_{0,i}^2)e^{-t\sqrt{4\sigma_i^2 + \lambda_{0,i}^2}}, \quad (16)$$

which depends on both $\sigma_i$ (singular values of the data) and $\lambda_{0,i}$ (level of initialization imbalance).

Table 2: Regimes where implicit acceleration is achieved (✓) or not achieved (✗) depending on the data singular values $\sigma$ and the level of imbalance $\lambda_0$ of the initial conditions (see (16)).

| | Balanced $\lambda_0 \approx 0$ | Imbalanced $\lambda_0 \gg 0$ |
|---|---|---|
| small $\sigma$ | ✗ | ✓ |
| large $\sigma$ | ✓ | ✓ |

It is immediate to see that over-parameterization leads to implicit acceleration when $4\sigma_i^2 + \lambda_{0,i}^2 > 1$. As illustrated in

Table 2 and previously observed in the literature (Saxe et al., 2014), acceleration is possible under a balanced initialization i.e., $\lambda_{0,i} \approx 0$ when the singular values are sufficiently large. This is similar to the symmetric case, which is not surprising since a symmetric factorization is by construction balanced. A new phenomenon that emerges from our analysis is that acceleration can also be achieved by increasing the level of imbalance, and that acceleration can always be achieved regardless of the data when $|\lambda_{0,i}| > 1$. Moreover, a faster convergence can be achieved by using a more imbalanced initialization.

## 4    Dynamics of Gradient Flow for Asymmetric Matrix Factorization without Spectral Initialization

We now relax the assumption of spectral initialization (Definition 3). Defining the quantities

$$R(t) \equiv \begin{bmatrix} \bar{U}(t) \\ \bar{V}(t) \end{bmatrix}, \qquad S \equiv \begin{bmatrix} 0 & \Sigma \\ \Sigma^T & 0 \end{bmatrix}, \qquad \bar{S} \equiv \frac{1}{2}\begin{bmatrix} I_m & 0 \\ 0 & -I_n \end{bmatrix}, \qquad (17)$$

one can immediately obtain from (13) and (14) the Riccati-like differential equation

$$\dot{R} = SR - \tfrac{1}{2}RR^T R + \bar{S}R\Lambda_{\mathcal{Q}_0}, \qquad (18)$$

where from (14) we conclude that $2R_0^T \bar{S} R_0 = \Lambda_{\mathcal{Q}_0}$ with $R(0) \equiv R_0$. However, in general, one cannot go back from (18) to (13) unless the conservation law (14) is explicitly imposed for all times $t$. The natural question is then, when are they equivalent? Our next result provides the answer and additionally reveals an interesting relation between (18) and a matrix factorization problem with explicit regularization (proof in Appendix F).

**Proposition 4 (Explicit regularization).** *The differential equation* (18) *is equivalent to*

$$\begin{aligned} \dot{\bar{U}} &= (\Sigma - \bar{U}\bar{V}^T)\bar{V} - \tfrac{1}{2}\bar{U}(\bar{U}^T\bar{U} - \bar{V}^T\bar{V} - \Lambda_{\mathcal{Q}_0}), \\ \dot{\bar{V}} &= (\Sigma - \bar{U}\bar{V}^T)^T\bar{U} + \tfrac{1}{2}\bar{V}(\bar{U}^T\bar{U} - \bar{V}^T\bar{V} - \Lambda_{\mathcal{Q}_0}). \end{aligned} \qquad (19)$$

*This system corresponds to the dynamics of gradient flow applied to the regularized problem*

$$\min_{\bar{U},\bar{V}} \left\{ \tfrac{1}{2}||\Sigma - \bar{U}\bar{V}^T||_F^2 + \tfrac{1}{8}||\bar{U}^T\bar{U} - \bar{V}^T\bar{V} - \Lambda_{\mathcal{Q}_0}||_F^2 \right\}. \qquad (20)$$

*Moreover, if $\bar{\mathcal{Q}}(t) \equiv \bar{U}^T(t)\bar{U}(t) - \bar{V}^T(t)\bar{V}(t)$ obeys $\bar{\mathcal{Q}}(t_0) = \Lambda_{\mathcal{Q}_0}$ at some $t = t_0$, then $\bar{\mathcal{Q}}(t) = \Lambda_{\mathcal{Q}_0}$ for all $t \geq t_0$. In particular, if we initialize* (19)—*or equivalently* (18)—*such that $\bar{\mathcal{Q}}(0) = 2R_0^T\bar{S}R_0 = \Lambda_{\mathcal{Q}_0}$, then the conservation law* (14) *holds true for all t and both the dynamics of* (19) *and* (13) *are the same.*

Some remarks are appropriate:

- A solution to (13) implies a solution to (19) because when (11) holds the 2nd terms on the RHS of (19) vanish, while the 1st terms are exactly (13). However, the converse is not necessarily true, unless (19) is initialized in the same way as (13). The above result relates *implicit acceleration* (as will be shown in Proposition 5) to an *explicit regularization*; namely one can either select a particular initialization and solve an unregularized problem, or start at an arbirary initialization and explicitly regularize.

- Note the specific weight of 1/8 in (20) is special: If one replaces 1/8 by some constant $\alpha > 0$, the gradient flow dynamics, i.e. the analog of (19), will not be equivalent to (18). Note also problem (20) also appeared in (Du et al., 2018a), but without such connections.

Equation (18) is reminiscent of a Riccati differential equation due to the cubic term in $R$ (similar to the gradient flow in the symmetric case) but we believe that, in general, it cannot be solved exactly due to the last term. However, it can be solved exactly in a particular case (proof in Appendix E).

**Proposition 5 (Exact solution and convergence rate).** *If $\Lambda_{\mathcal{Q}_0} = \lambda_0 I_k$ for some constant $\lambda_0$, then the differential equation* (18) *reduces to the following equation with a closed form solution for $RR^T$*

$$\dot{R} = \tilde{S}R - \tfrac{1}{2}RR^T R, \qquad R(t)R^T(t) = e^{t\tilde{S}}R_0R_0^T\left(I + \tfrac{1}{2}\tilde{S}^{-1}(e^{2t\tilde{S}} - I)R_0R_0^T\right)^{-1}e^{t\tilde{S}}, \qquad (21)$$

where $\tilde{S} \equiv S + \lambda_0 \bar{S} = \Phi\tilde{\Sigma}\Phi^T$, $R_0 \equiv R(0)$. Moreover, if $\tilde{S}$ and $I + \frac{1}{2}\hat{S}^{-1}(R_0 R_0^T - R_\star R_\star^T)$ are invertible then $R(t)R^T(t)$ converges exponentially to $R_\star R_\star^T$, defined as the projection of the matrix $2\tilde{S}$ on the PSD cone, $\hat{S} = \Phi|\tilde{\Sigma}|\Phi^T$. More precisely, if $Y$ is a square matrix the convergence rate is $O\big(e^{-t\sqrt{4\sigma_{\min}^2 + \lambda_0^2}}\big)$, where $\sigma_{min}$ is the smallest eigenvalue of $Y$, and otherwise the rate is $O(e^{-|\lambda_0|t})$.

Note that (21) is the gradient flow for the symmetric factorization problem $\min_R \left\{ \frac{1}{8}||2\tilde{S} - RR^T||_F^2 \right\}$. The particular case $\Lambda_{Q_0} = \lambda_0 I_k$ is mathematically interesting because it is amenable to an analytical treatment. However, it may not be realizable in practice because the conserved quantity $Q_0$ (or $\Lambda_{Q_0}$) must have low rank, i.e.,

$$\text{rank}(Q_0) = \text{rank}(U_0^T U_0 - V_0^T V_0) \le \text{rank}(U_0^T U_0) + \text{rank}(V_0^T V_0) \le m + n. \qquad (22)$$

Since $\text{rank}(\lambda_0 I_k) = k$, choosing $\bar{U}_0$ and $\bar{V}_0$ such that $\bar{U}_0^T \bar{U}_0 - \bar{V}_0^T \bar{V}_0 = \lambda_0 I_k$ is not generally possible in an over-parameterized setting with $k > m+n$. On the other hand, the choice $\Lambda_{Q_0} = \lambda_0 I_k$ does not present a problem if we consider the system (19) where we have the freedom to choose any initialization. The experiments in Section 5 illustrate that (19), or equivalently (21), is actually enough to capture the general behaviour of (13).

## 5 Numerical Experiments

**Imbalanced initialization.** We now provide numerical evidence supporting our theoretical results. First, we generate a random matrix $Y$ with $Y_{ij} \sim \mathcal{N}(0,1)$, and set $m = 5$, $n = 10$ and $k = 50$. We approximate the dynamics of gradient flow for one-layer and two-layer linear models by using gradient descent with a fixed small step size $\eta = 10^{-3}$ (smaller step sizes did not lead to a discernible change). We evaluate the *reconstruction error* $\|Y - X(t)\|_F / \|Y\|_F$, where $X(t) = U(t)V^T(t)$, and compare the evolution of the singular values of $X(t)$. We consider Gaussian initializations, i.e., $U_0$ and $V_0$ have entries $\sim \mathcal{N}(0, \sigma^2)$, where $\sigma$ is varied to obtain different degrees of imbalance. In order for the two models to start from the same state, we choose $X(0) = U_0 V_0^T$ for the one-layer case. The results are shown in Fig. 1. From our theoretical analysis, we expect a different behaviour for the convergence rate depending on whether the initialization is balanced or imbalanced, i.e. whether $\|Q\|_F = \|Q_0\|_F \equiv \|U_0^T U_0 - V_0^T V_0\|_F$ is small or large, respectively. When it is very small (Fig. 1a), the strength of the singular value dominates and we expect the components to be learned sequentially from the largest to the smallest, which agrees with the findings in (Saxe et al., 2019; Gidel et al., 2019). As we make the weights more imbalanced (Fig. 1b) the singular values are learned faster, even the smaller ones. Finally, as

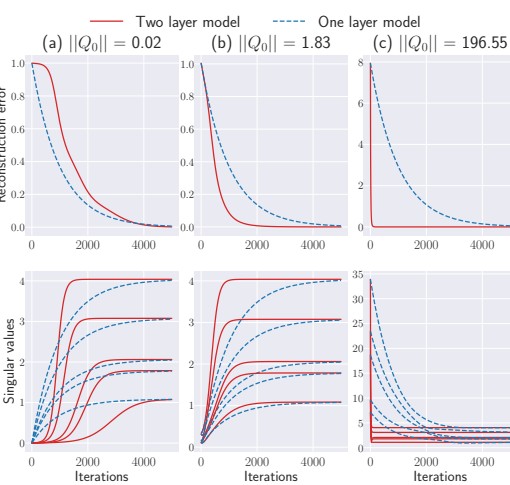

Figure 1: *Top row:* reconstruction error for one- vs. two-layer linear models. *Bottom row:* evolution of singular values of the solution. From left to right we use $\sigma = 10^{-2}$, $\sigma = 10^{-1}$, and $\sigma = 1$ respectively.

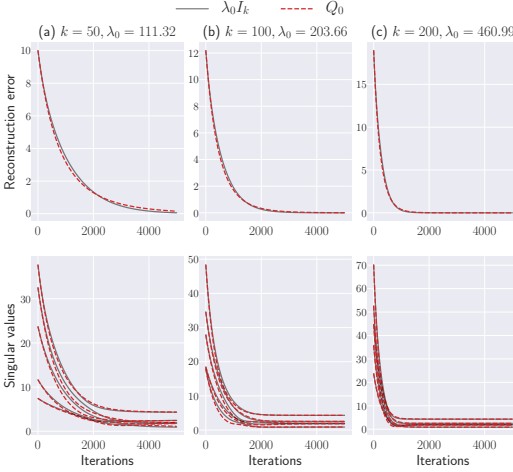

Figure 2: *Top row:* reconstruction error for the asymmetric factorization dynamics without regularization in (13) and (14) and general $Q_0$ (red dashed line), versus the regularized dynamics in (19) with a diagonal $\Lambda_{Q_0} = \lambda_0 I_k$ (black solid line). *Bottom row:* evolution of the corresponding singular values of the solutions. From left to right we set $k=50$, $k=100$, and $k=200$, respectively.

$\|\mathcal{Q}\|_F$ becomes very large, the implicit acceleration becomes more prominent and the solution of the factorized problem converges significantly faster (Fig. 1c). In other words, these numerical results are consistent with Proposition 2 and Proposition 5.

$\mathbf{\Lambda_{\mathcal{Q}_0} = \lambda_0 I}$ **is general enough.** Since Proposition 5 contains the case where an exact solution is available, we want to investigate whether this is general enough to capture the qualitative behaviour of the general problem, i.e. the general dynamics of (13). To avoid confusion, let us refer to $\bar{U}^I$ and $\bar{V}^I$ as the variables of (19), as well as $\Lambda_{\mathcal{Q}_0}^I \equiv \lambda_0 I_k$; here $I$ stands for "identity." The variables $\bar{U}$ and $\bar{V}$ refer to the original dynamical system (13), with its conserved quantity $\Lambda_{\mathcal{Q}}$ completely fixed by the initial conditions; see (14). We want to show that it is possible to find an "optimal" $\lambda_0 \in \mathbb{R}$ such that both cases have very close dynamics. Hence, we initialize $U_0$ and $V_0$ (and thus equivalently $\bar{U}_0$ and $\bar{V}_0$) with entries sampled from $\mathcal{N}(0, \sigma^2)$, and we fix $\sigma = 1$. The same initial condition is also used for (19), i.e. $\bar{U}_0^I = \bar{U}_0$ and $\bar{V}_0^I = \bar{V}_0$. We set $\eta = 10^{-5}$, $Y \sim \mathcal{N}(0,1)$, $m = 5$, $n = 10$ and vary $k$. Thus, we look for a $\lambda_0$ that minimizes the error $\|X^I(t) - X(t)\|_F$. In Fig. 2 we illustrate that, indeed, this can be done. Note that in these three cases the evolution of both dynamical systems are nearly indistinguishable.

**Extension to nonlinear networks.** Our analysis so far has shown that the acceleration phenomenon is a result of imbalance and is induced through a conservation law for which the definition is expected to change when introducing nonlinearities. In fact, both the architecture of the network and the objective function should affect the invariances. As such, conducting a full analysis of the symmetries and conservation laws for more complex nonlinear networks is necessary to characterize the implicit bias in such cases, which we leave for future work.

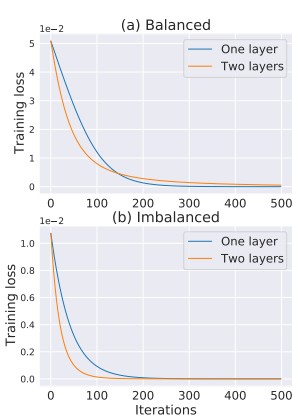

Nonetheless, we provide numerical evidence in a more realistic setting than the one of linear networks, where we only add a nonlinearity (sigmoid) to the final layer in order to preserve some of the structure of the linear model. We train the two networks (one layer vs two layers) on synthetic data, i.e. we compare the dynamics of gradient descent for the objectives: $\ell_1(W) = \frac{1}{2}\|Y - \phi(XW)\|_F^2$ and $\ell_2(U, V) = \frac{1}{2}\|Y - \phi(XUV^T)\|_F^2$ where $\phi$ is the sigmoid function, $X \in \mathbb{R}^{n \times d}$ and $Y \in \mathbb{R}^{n \times d}$ represent the training samples and labels respectively ($n = 10^3, d = 10$), $W \in \mathbb{R}^{d \times d}, U, V \in \mathbb{R}^{n \times k}$ are the weight matrices and the width is $k = 100$. We generated the matrices $W^*$ and $X$ with entries drawn from $\mathcal{N}(0,1)$ and $Y = \phi(XW^*) + \epsilon$ where $\epsilon \sim 10^{-3}\mathcal{N}(0,I)$. Our results shown in figure 3 interestingly suggest that our conclusions about the role of imbalance hold in this case as well.

Figure 3: Evolution of the training loss for nonlinear one-layer and two-layer models. *Top row*: $\|Q_0\|_2 = 0$. *Bottom row*: $\|Q_0\|_2 = 4.6$. Initial weights are drawn from a normal distribution $\mathcal{N}(0, 10^{-1})$.

## 6 CONCLUSION

We considered the gradient flow dynamics for two-layer linear neural networks, providing an analytical treatment to a great level of detail. Our results establish a detailed characterization of the "implicit acceleration" phenomenon in this case, without assuming balanced or vanishingly small initializations, which so far have been present in all prior work in this vein. More specifically, our analysis shows that the implicit acceleration of gradient flow is a consequence of an emerging rotational symmetry induced by over-parameterization and giving rise to several conservation laws that constrain the dynamics to follow specific trajectories. Moreover, conserved quantities are completely fixed by the initialization, which has profound effects on the convergence rate. Although our analysis focuses on the simple case of linear networks, it reveals a potential key to understanding implicit bias which lies in the conservation laws that arise from the symmetries of the problem. These symmetries depend on the network architecture, objective, optimization algorithm, and they constrain the dynamics to an invariant manifold that encapsulates the implicit regularization and acceleration effects. Understanding this in more complex models may thus be reduced to finding dynamical invariants, for which our results provide a foundational starting point.

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

# A    Solution to the Matrix Riccati Differential Equation

Here we prove Proposition 1.

First, we note that a general matrix Riccati differential equation takes the form

$$\dot{P}(t) = AP(t) + P(t)A^T - P(t)RP(t) + Q, \qquad P(0) = P_0, \tag{23}$$

where $P(t) \in \mathbb{R}^{n \times n}$, and $A, R, Q, P_0 \in \mathbb{R}^{n \times n}$ are constant matrices. Associated to (23) one has the linear system

$$\begin{bmatrix} \dot{X}_1(t) \\ \dot{X}_2(t) \end{bmatrix} = \begin{bmatrix} -A^T & R \\ Q & A \end{bmatrix} \begin{bmatrix} X_1(t) \\ X_2(t) \end{bmatrix}, \qquad \begin{bmatrix} X_1(0) \\ X_2(0) \end{bmatrix} = \begin{bmatrix} I_n \\ P_0 \end{bmatrix}. \tag{24}$$

The closed form solution of (23) follows as a consequence of lemma 6 below.

**Lemma 6** ((Sasagawa, 1982)). *Consider the two initial value problems (23) and (24). We have:*

- *The initial value problem (23) has a solution in the interval $[0, t_1]$ if and only if the matrix $X_1(t)$ in the solution of the linear differential equation (24) is invertible for all $t \in [0, t_1)$. Moreover, the solution to (23) is unique and given by*

$$P(t) = X_2(t)X_1(t)^{-1}. \tag{25}$$

- *Let $\bar{P}$ be a solution to the algebraic Riccati equation (ARE)*

$$AP + PA^T - PRP + Q = 0. \tag{26}$$

*Then the solution of (24) is given by (27) below, where $\tilde{A} = A - \bar{P}R$ and $\hat{A} = A^T - R\bar{P}$:*

$$\begin{bmatrix} X_1(t) \\ X_2(t) \end{bmatrix} = \begin{bmatrix} e^{-t\hat{A}} + \left( \int_0^t ds\, e^{-(t-s)\hat{A}} R\, e^{s\tilde{A}} \right) (P_0 - \bar{P}) \\ \bar{P}e^{-t\hat{A}} + \bar{P} \left( \int_0^t ds\, e^{-(t-s)\hat{A}} R\, e^{s\tilde{A}} \right) (P_0 - \bar{P}) + e^{t\tilde{A}}(P_0 - \bar{P}) \end{bmatrix}. \tag{27}$$

Now, we apply Lemma 6 to the Riccati equation induced by a symmetric factorization, namely,

$$\dot{X}(t) = 2X(t)Y + 2YX(t) - 4X(t)^2. \tag{28}$$

The associated linear system is

$$\begin{bmatrix} \dot{X}_1(t) \\ \dot{X}_2(t) \end{bmatrix} = \begin{bmatrix} -2Y & 4I \\ 0 & 2Y \end{bmatrix} \begin{bmatrix} X_1(t) \\ X_2(t) \end{bmatrix}, \qquad \begin{bmatrix} X_1(0) \\ X_2(0) \end{bmatrix} = \begin{bmatrix} I_n \\ X_0 \end{bmatrix}. \tag{29}$$

and the algebraic Riccati equation is given by

$$XY + YX - 2X^2 = 0. \tag{30}$$

This equation admits the trivial solution $X = 0$. Thus, one can verify that as long as the given matrix $Y$ is invertible we have

$$\begin{bmatrix} X_1(t) \\ X_2(t) \end{bmatrix} = \begin{bmatrix} e^{-2tY} + e^{-2tY}Y^{-1}(e^{4Yt} - I)X_0 \\ e^{2tY}X_0 \end{bmatrix}. \tag{31}$$

Therefore, the unique solution to (5) is explicitly given by

$$X(t) = X_2(t)X_1(t)^{-1} = e^{2tY}X_0 \left( I + Y^{-1}(e^{4tY} - I)X_0 \right)^{-1} e^{2tY}, \tag{32}$$

as long as the matrix between parenthesis is invertible. Note that $Y^{-1}(e^{4tY} - I)$ is always positive semidefinite for $t \geq 0$. Thus, for all $X_0 \succeq 0$, the matrix $I + Y^{-1}(e^{4tY} - I)X_0$ is invertible and the solution of the Riccati equation is well defined for all $t \geq 0$.

## B    Proof of Corollary 1

The first part follows trivially by substituting in (6) and verifying the invertibility of the matrix in parenthesis. For the second part, note from (7) that if $\sigma_i > 0$, then $\sigma_i(t) \to \sigma_i$ as $t \to \infty$ at a rate $O(e^{-4t\sigma_i})$, and if $\sigma_i < 0$, then $\sigma_i(t) \to 0$ at a rate $O(e^{4t\sigma_i})$. Therefore, $\Sigma(t) \to \max(\Sigma, 0)$ and $X(t) \to \Phi \max(\Sigma, 0)\Phi^T$ at a rate $O(e^{-4t\sigma_{\min}(Y)})$, as claimed.

## C    Rate of Convergence in the Symmetric Case

In this section, we show that the solution of the Riccati equation $\dot{X}(t) = 2YX(t) + 2X(t)Y - 4X(t)^2$ converges exponentially to $\tilde{Y}$ at a rate $O(e^{-4t\sigma_{\min}(Y)})$, where $\sigma_{\min}(Y) = \min_i |\sigma_i|$ is the smallest eigenvalue of $Y$ in magnitude. We already know that if $X(t)$ converges to $X_*$, then

- $X_*$ is positive semidefinite if $X_0 \succeq 0$ and $\mathrm{rank}(X_*) \leq \mathrm{rank}(X_0)$,
- $X_*$ is a solution to the algebraic Riccati equation $2YX + 2XY - 4X^2 = 0$.

Our proof of proposition 2 is inspired by the proofs in (Molinari, 1977; Callier et al., 1992; 1994) which studied the solution of the Riccati equation under a more general setting. The strategy will be to show that the algebraic Riccati equation has a unique PSD solution $X_+$ such that the eigenvalues of $\tilde{A} = \hat{A} = 2Y - 4X_+$ have negative real parts. Such solution is usually referred to as the strong solution or stabilizing solution in optimal control literature because it is the only solution of the algebraic Riccati equation such that the matrix $\tilde{A}$ is exponentially stable, i.e. $\exp(t\tilde{A})$ converges to 0. The stability of the matrix $\tilde{A}$ is important because it appears in the solution of the Riccati equation as can be observed in (27).

We start by proving that $\tilde{Y}$ is a solution to the algebraic Riccati equation, i.e. it is a critical point of the problem.

Due to the symmetric and positive semidefinite nature of the matrices X and Y, in our case the algebraic equation (30) can be reduced to

$$X(X - Y) = 0. \tag{33}$$

For $X = \tilde{Y}$, we thus have

$$
\begin{aligned}
\tilde{Y}(\tilde{Y} - Y) &= \left( \sum_{i=1}^{m} \max\{\sigma_i, 0\}\phi_i\phi_i^T \right)\left( \sum_{i=1}^{m} (\max\{\sigma_i, 0\} - \sigma_i)\phi_i\phi_i^T \right) \\
&= \left( \sum_{i}^{m} \max\{\sigma_i, 0\}\phi_i\phi_i^T \right)\left( \sum_{i=1}^{m} \min\{\sigma_i, 0\}\phi_i\phi_i^T \right).
\end{aligned}
\tag{34}
$$

The first sum contains only the positive eigenvalues while the second sum contains only the negative ones. Therefore, no eigenvalue will appear in both sums and using the orthogonality of the vectors $\phi_i$ we conclude that $\tilde{Y}(\tilde{Y} - Y) = 0$.

Next, we prove that $\bar{P} = \tilde{Y}$ is the unique symmetric positive semidefinite solution of the algebraic Riccati equation such that the eigenvalues of $\tilde{A} = A - R\bar{P}$ have negative real parts. Note that $\tilde{A} = 2Y - 4\tilde{Y} = -2\hat{Y}$ where $\hat{Y} = \sum_{i=1}^{m} |\sigma_i|\phi_i\phi_i^T \succ 0$.

We proceed by contradiction. Let $X_2$ be a PSD solution of the (ARE) such that the eigenvalues of $\tilde{A}_2 = 2Y - 4X_2$ have negative real parts and $X_2 \neq \tilde{Y}$.

Now consider $\Delta = \tilde{Y} - X_2$. By a straightforward calculation, we can show that $\Delta$ is a solution of

$$\tilde{A}\Delta + \Delta\tilde{A} + 4\Delta^2 = 0. \tag{35}$$

Since $\Delta$ is not necessarily invertible, we consider a basis such that

$$\Delta = Z \begin{bmatrix} 0 & 0 \\ 0 & D \end{bmatrix} Z^T = Z\tilde{\Delta}Z^T, \tag{36}$$

where $D$ is invertible. Note that our proof holds and is more trivial if $\Delta$ is invertible. We write

$$\tilde{A} = Z \begin{bmatrix} W_1 & W_2 \\ W_3 & W_4 \end{bmatrix} Z^T = Z\bar{A}Z^T \tag{37}$$

After the change of basis, equation (35) becomes

$$\bar{A}\tilde{\Delta} + \tilde{\Delta}\bar{A} + 4\tilde{\Delta}^2 = 0. \tag{38}$$

From (35), we can deduce the following for the block matrices:

$$W_2 = 0,$$
$$W_3 = 0,$$
$$DW_4 + W_4 D + 4D^2 = 0.$$

Note that the last equation is similar to equation (35) for the invertible block $D$. Moreover, in the new basis, $\tilde{A}$ is a block diagonal matrix. Using the change of variable $T = D^{-1}$, we obtain the Lyapunov equation

$$TW_4 + W_4 T + 4I = 0. \tag{39}$$

Since $\tilde{A} = -2\hat{Y} \prec 0$ is invertible, the block $W_4$ is also invertible and its eigenvalues are a subset of the eigenvalues of $-2\hat{Y}$. As a result, the Lyapunov equation has the unique trivial solution $T = -2W_4^{-1} \succ 0$, therefore $D = -\frac{1}{2}W_4 \succ 0$.

Using the solution of (35), we can obtain a new derivation for $\tilde{A}_2$ as follows

$$\tilde{A}_2 = 2Y - 4X_2 = 2Y - 4\tilde{Y} + 4\Delta$$

$$= \tilde{A} + 4\Delta = Z \begin{bmatrix} W_1 & 0 \\ 0 & -W_4 \end{bmatrix} Z^T$$

with $W_1 \prec 0$ and $-W_4 \succ 0$. This contradicts with the initial assumption on the eigenvalues of $\tilde{A}_2$ having negative real parts.

Now we can use Lemma 6 with $\bar{P} = \tilde{Y}$ to obtain a new expression of the solution. We have $\tilde{A} = \hat{A} = -2\hat{Y}$ and

$$\begin{bmatrix} X_1(t) \\ X_2(t) \end{bmatrix} = \begin{bmatrix} e^{-2t\hat{Y}}\left[I + \hat{Y}^{-1}(I - e^{-4\hat{Y}t})(X_0 - \tilde{Y})\right] \\ \tilde{Y}e^{-2t\hat{Y}}\left[I + \hat{Y}^{-1}(I - e^{-4\hat{Y}t})(X_0 - \tilde{Y})\right] + e^{2t\hat{Y}}(X_0 - \tilde{Y}) \end{bmatrix}. \tag{40}$$

Therefore, the solution of the Riccati equation is also given by

$$X(t) = X_2(t)X_1(t)^{-1} = \tilde{Y} + e^{-2t\hat{Y}}(X_0 - \tilde{Y})\left[I + \hat{Y}^{-1}(I - e^{-4\hat{Y}t})(X_0 - \tilde{Y})\right]^{-1}e^{-2t\hat{Y}} \tag{41}$$

Note that the inverse exists for $t \geq 0$ when $X_0 \succeq 0$ because we have proven the existence of the solution using the previous expression (32) (see lemma 2 in (Sasagawa, 1982) for a detailed proof).

We introduce the function $H(t) = (X_0 - \tilde{Y})\left[I + \hat{Y}^{-1}(I - e^{-4\hat{Y}t})(X_0 - \tilde{Y})\right]^{-1}$. Thus,

$$X(t) - \tilde{Y} = e^{-2t\hat{Y}}H(t)e^{-2t\hat{Y}} \tag{42}$$

The function $H$ has the following properties for $t \geq 0$:

- It is decreasing:

$$\frac{dH(t)}{dt} = -4H(t)e^{-4\hat{Y}t}H(t) \leq 0 \tag{43}$$

- If $I + \hat{Y}(X_0 - \tilde{Y})$ is invertible then

$$\lim_{t \to \infty} H(t) = (X_0 - \tilde{Y})[I + \hat{Y}(X_0 - \tilde{Y})]^{-1} = \tilde{H}. \tag{44}$$

- $H$ is bounded on $\mathbb{R}_+$;

$$\tilde{H} \leq H(t) \leq H(0) = X_0 - \tilde{Y}. \tag{45}$$

Therefore, we can conclude that there exists a constant $C > 0$ such that

$$\|X(t) - \tilde{Y}\|_F \leq Ce^{-4\sigma_{min}t}, \tag{46}$$

where $\sigma_{min} = \min\{|\sigma_i|\}$ is the smallest eigenvalue of $Y$ in absolute value.

# D    Closed form solution for the asymmetric case under spectral initialization

Under the spectral initialization, $\bar{U}_0$ and $\bar{V}_0$ are diagonal, hence so are $\dot{\bar{U}}(0)$ and $\dot{\bar{V}}(0)$. As a result, $\dot{\bar{U}}(t)$ and $\dot{\bar{V}}(t)$ remain diagonal for all $t \geq 0$, because the components of (13) can be decoupled and the evolution of (13) will induce no change in the off-diagonal elements. To see this, observe that

$$\dot{\bar{U}}_{ii} = (\sigma_i - \bar{U}_{ii}\bar{V}_{ii})\bar{V}_{ii}, \qquad \dot{\bar{V}}_{ii} = (\sigma_i - \bar{U}_{ii}\bar{V}_{ii})\bar{U}_{ii} \qquad (1 \leq i \leq m), \tag{47}$$

whereas the off-diagonal terms obey $\dot{\bar{U}}_{ij} = 0$ and $\dot{\bar{V}}_{ij} = 0$ ($i \neq j$). Thus (47) describes the evolution of the *singular values of the solution*. This decouples the problem into a set of independent one-dimensional equations, therefore it suffices to consider the scalar system

$$\dot{\bar{u}} = (\sigma - \bar{u}\bar{v})\bar{v}, \qquad \dot{\bar{v}} = (\sigma - \bar{u}\bar{v})\bar{u}, \tag{48}$$

where we drop the index $i = 1, \dots, m$ for simplicity. In (Saxe et al., 2019) there is a strong assumption, i.e. $\bar{u}_{ii}(0) = \bar{v}_{ii}(0)$ for all $i$ which is a *balanced initialization* (Definition 2). Here we solve (47) without such an assumption. From (48), it is immediate that the conservation law (11) becomes $\frac{d}{dt}\left(\bar{u}^2 - \bar{v}^2\right) = 0$. Trajectories $(\bar{u}(t), \bar{v}(t))$ are thus constrained to lie on hyperbolas defined by

$$\bar{u}^2(t) - \bar{v}^2(t) = \bar{u}_0^2 - \bar{v}_0^2 = \lambda_0 = \text{const.} \tag{49}$$

Since we are mostly interested in the behavior of the product $x(t) = \bar{u}(t)\bar{v}(t)$ by making explicit use of the conservation law (49), specifically $\lambda_0^2 = \bar{u}^4 - 2\bar{u}^2\bar{v}^2 + \bar{v}^4 = \bar{u}^4 + \bar{v}^4 - 2x^2$, and $(\bar{u}^2 + \bar{v}^2)^2 = \lambda_0^2 + 4x^2$, we obtain the following first-order differential equation for $x \equiv x(t)$:

$$\dot{x} = (\sigma - \bar{u}\bar{v})\bar{v}^2 + (\sigma - \bar{u}\bar{v})\bar{u}^2 = (\sigma - \bar{u}\bar{v})\sqrt{(\bar{v}^2 + \bar{u}^2)^2} = 2(\sigma - x)\sqrt{x^2 + \lambda_0^2/4}. \tag{50}$$

Even though this is a nonlinear differential equation, it is separable, thus integrating both sides yields precisely (15) (we restore $i$ and $x \to \sigma_i$ represents the corresponding component associated with singular value $\sigma_i$ and conserved quantity $\lambda_{0,i}$), where $C > 0$ is a constant:

$$C = \frac{\sqrt{4\sigma^2\lambda_0^2 + 16\sigma^2 x_0^2 + \lambda_0^4 + 4x_0^2\lambda_0^2} - 4\sigma x_0 - \lambda_0^2}{4\lambda_0^2(\sigma - x_0)}. \tag{51}$$

Above, only $m$ out of $k \geq m$ conserved quantities are used. Hence, there is degeneracy in the solution and only $m$ effective degrees of freedom regardless how large $k$ is. Note that if $k < m$ (under-parameterized case) then (47) becomes under-determined.

# E    Rate of Convergence in the Asymmetric Case

Here we prove Proposition 5. First, note that equation $\dot{R} = \tilde{S}R - \frac{1}{2}RR^TR$ follows directly from $\dot{R} = SR - \frac{1}{2}RR^TR + \bar{S}R\Lambda_{\mathcal{Q}_0}$ when $\Lambda_{\mathcal{Q}_0} = \lambda_0 I$. This leads to a Riccati differential equation for $P(t) = R(t)R^T(t)$:

$$\begin{aligned} \frac{dP(t)}{dt} &= \dot{R}R^T + R\dot{R}^T = \left(\tilde{S}R - \frac{1}{2}RR^TR\right)R^T + R\left(\tilde{S}R - \frac{1}{2}RR^TR\right)^T \\ &= \tilde{S}P(t) + P(t)\tilde{S} - (P(t))^2 \end{aligned} \tag{52}$$

with initial conditions $P(0) = R_0R_0^T$. By the arguments used in Appendix A, the exact solution of the above equation is given by;

$$\left(RR^T\right)(t) = e^{t\tilde{S}}R_0R_0^T\left(I + \frac{1}{2}\tilde{S}^{-1}(e^{2t\tilde{S}} - I)R_0R_0^T\right)^{-1}e^{t\tilde{S}}. \tag{53}$$

One can show that $RR^T$ converges exponentially to the matrix $R_\star R_\star^T$; defined as the projection of $2\tilde{S}$ on the positive semidefinte cone; this is derived by the same arguments leading to Proposition 2 (see Appendix C). The convergence rate depends on the eigenvalues

of $2\tilde{S}$ which can be determined using Schur's complement formula. For a block matrix $M$ one has

$$M = \begin{bmatrix} A & B \\ C & D \end{bmatrix} \implies \det(M) = \det(D)\det(A - BD^{-1}C). \tag{54}$$

Thus, using the above formula for $\tilde{S} - \lambda I = \begin{bmatrix} (\lambda_0/2 - \lambda)I_m & \Sigma \\ \Sigma^T & -(\lambda_0/2 + \lambda +)I_n \end{bmatrix}$ we have

$$\det(\tilde{S} - \lambda I) = (-1)^{m+n}(\lambda_0/2 + \lambda)^{n-m}\prod_{i=1}^{m}\left(\sigma_i^2 + \lambda_0^2/4 - \lambda^2\right). \tag{55}$$

Therefore, the eigenvalues of $\tilde{S}$ are:

1. $2m$ eigenvalues $\pm\tilde{s}_i$ where $\tilde{s}_i = \frac{1}{2}\sqrt{4\sigma_i^2 + \lambda_0^2}$ $(i = 1, \ldots, m)$.
2. The eigenvalue $-\lambda_0/2$ of multiplicity at least $n - m$.

In general the smallest magnitude eigenvalue will be $|\lambda_0|/2$. However, if $Y$ is a square matrix then only the eigenvalues $\pm\tilde{s}_i$ above will be present.

## F    From Implicit Acceleration to Explicit Regularization

We now prove Proposition 4. Replacing (17) into (18) immediately yields (19). It is also straightforward to verify that applying the gradient flow, $\frac{d}{dt}\bar{U} = -\nabla_{\bar{U}}\ell$ and $\frac{d}{dt}\bar{V} = -\nabla_{\bar{V}}\ell$, with $\ell$ being the objective function in (20), yields (19). Next, consider the formal Taylor series

$$\bar{\mathcal{Q}}(t) = \bar{\mathcal{Q}}(t_0) + (t - t_0)\dot{\bar{\mathcal{Q}}}(t_0) + \frac{1}{2}(t - t_0)^2\ddot{\bar{\mathcal{Q}}}(t_0) + \cdots. \tag{56}$$

Define $\mathcal{P}(t) \equiv \bar{U}^T(t)\bar{U}(t) + \bar{V}^T(t)\bar{V}(t)$. From (19) one obtains

$$\frac{d}{dt}\bar{\mathcal{Q}}(t) = -\frac{1}{2}(\bar{\mathcal{Q}}(t) - \Lambda_{\mathcal{Q}_0})\mathcal{P}(t) - \frac{1}{2}\mathcal{P}(t)(\bar{\mathcal{Q}}(t) - \Lambda_{\mathcal{Q}_0}). \tag{57}$$

Since this is "linear" in $(\bar{\mathcal{Q}} - \Lambda_{\mathcal{Q}_0})$, higher order derivatives take the form

$$\frac{d^n}{dt^n}\bar{\mathcal{Q}}(t) = \sum_i \mathcal{Z}_i(t)\left(\bar{\mathcal{Q}}(t) - \Lambda_{\mathcal{Q}_0}\right)\mathcal{W}_i(t) \tag{58}$$

where the functions $\mathcal{Z}_i$'s and $\mathcal{W}_i$'s contain a sum of powers and time derivatives of $\mathcal{P}(t)$. For instance, the second order derivative yields

$$\ddot{\bar{\mathcal{Q}}} = \frac{d^2}{dt^2}\bar{\mathcal{Q}} = -\frac{1}{2}\dot{\mathcal{Q}}(t)\mathcal{P}(t) - \frac{1}{2}(\mathcal{Q} - \Lambda_{\mathcal{Q}_0})\dot{\mathcal{P}}(t) - \frac{1}{2}\mathcal{P}(t)\dot{\mathcal{Q}}(t) - \frac{1}{2}\dot{\mathcal{P}}(t)(\mathcal{Q} - \Lambda_{\mathcal{Q}_0})$$
$$= -\frac{1}{4}(\mathcal{Q} - \Lambda_{\mathcal{Q}_0})(\mathcal{P}^2 + 2\dot{\mathcal{P}}) - \frac{1}{4}(\mathcal{P}^2 + 2\dot{\mathcal{P}})(\mathcal{Q} - \Lambda_{\mathcal{Q}_0}) - \frac{1}{2}\mathcal{P}(\mathcal{Q} - \Lambda_{\mathcal{Q}_0})\mathcal{P}.$$

Therefore, if $\bar{\mathcal{Q}}(t_0) = \Lambda_{\mathcal{Q}_0}$ at $t = t_0$ then all derivatives (58) vanish identically. As a consequence, the expansion (56) implies $\bar{\mathcal{Q}}(t) = \bar{\mathcal{Q}}(t_0) = \Lambda_{\mathcal{Q}_0}$ for any other $t \geq t_0$ as well.

## G    On matching rates between Gradient Flow and Gradient Descent

We provide an explicit example to illustrate why studying the continuous-time dynamics of the gradient flow is expected to reproduce the behaviour of its discretization, i.e. gradient descent. For the sake of simplicity let us limit the discussion to the case considered in Section 2 and in the scalar case, $Y = \sigma \in \mathbb{R}$. What we would really like to do is to compare two different algorithms, namely gradient descent applied to problem (2) versus GD applied to the factorized problem (3). We thus have

$$X_{k+1} = X_k + \eta(\sigma - X_k) \tag{59}$$

versus

$$U_{k+1} = U_k + 2\eta(\sigma - U_k^2)U_k. \tag{60}$$

The respective continuous-time limits of these algorithms are

$$\dot{X} = (\sigma - X) \tag{61}$$

versus

$$\dot{U} = 2(\sigma - U^2)U, \tag{62}$$

for $X = X(t)$ and $U = U(t)$. The solution of (61) is given by $X(t) - \sigma = (X_0 - Y)e^{-t}$, yielding a rate $O(e^{-t})$. Let us introduce the perturbed variable $\tilde{X}_k \equiv X_k - \sigma$. Hence (59) readily gives $\tilde{X}_{k+1} = (1 - \eta)\tilde{X}_k$, thus a matching rate of $O(e^{-\eta k})$. (This example is trivial because both systems are linear.) Now let us move on to the more interesting nonlinear case. Consider (61) and let $X(t) \equiv U^2(t)$. Thus $\dot{X} = 4\sigma X - 4X^2$, whose solution is

$$X(t) = \frac{\sigma}{1 - ce^{-4\sigma t}} \approx \sigma - ce^{-4\sigma t}, \tag{63}$$

which implies a continuous-time rate of $O(e^{-4\sigma t})$. (Compare this with the last phrase in Corrolary 1.) Now let us see what happens for (60). Note that this is a complicated nonlinear recurrence relation. Fortunately, we can solve it approximately. By introducing $X_k \equiv U_k^2$ this recurrence becomes

$$X_{k+1} = X_k + 4\eta(\sigma - X_k)X_k + 4\eta^2(\sigma - X_k)^2 X_k. \tag{64}$$

Now we do the following trick. Let $\tilde{X}_k \equiv X_k - \sigma$ be a perturbation from equilibrium. For a small enough $\eta$ we can neglect terms of $O(\eta^2)$, hence $\tilde{X}_{k+1} \approx \tilde{X}_k - 4\eta\tilde{X}_k(\sigma + \tilde{X}_k) \leq (1 - 4\eta\sigma)\tilde{X}_k$. This implies that $\tilde{X}_k \to 0$, or equivalently $X_k \to \sigma$, at a discrete-time rate of $e^{-4\sigma\eta k}$. This matches the continuous-time rate.

It is not hard to see how such nonlinear recurrence relations quickly become intractable for more complicated problems. On the other hand, even though the continuous-time limit provided by the gradient flow consists of a nonlinear ODE, the analysis is much more feasible, besides introducing connections with interesting areas of mathematics such as Riccati differential equations.

