# OpenReview forum: "Implicit Acceleration of Gradient Flow in Overparameterized Linear Models"
_ICLR.cc/2021/Conference — Reject_

### Official Review · AnonReviewer3 · 2020-10-27
**Review for 'Implicit Acceleration of Gradient Flow in Overparameterized Linear Models'**

**Rating:** 6
**Confidence:** 3

**Review:**

This paper considers the gradient flow dynamic for two-player linear neural networks. In detail, it studies the implicit acceleration of gradient flow brought by overparameterization and shows the reason for implicit acceleration is the existence of conservation law. It studies the convergence for gradient flow under both balanced or imbalanced linear networks, and with spectral or non-spectral initialization. Compared with previous work, this work is the first to provide an explicit characterization of the gradient flow with respect to their eigenvalues. Experiment results suggest that such an implicit acceleration indeed exists.

Here are my detailed comments.

- Page 2, (2): why call (2) ‘a symmetric one-layer linear model’? Does that suggest $m = n$? The same issue holds for (3). Is $Y$ in (3) the same as that in (2)?

- An interesting observation is that under the spectral initialization, for the symmetric case, the convergence rate of the eigenvalues does not depend on the initial value of $X_0$ (it is $e^{-4t|\sigma_i|}$). However, for the asymmetric case, the convergence rate is $e^{-t\sqrt{4\sigma_i^2 + \lambda_0^2}}$, which explicitly depends on the initial matrix $X_0$. Can the authors make more comments about that?

- The ‘acceleration’ compared with original gradient flow over $X$ suggests that by a matrix factorization, the convergence of eigenvalues varies may be accelerated according to the eigenvalues over the data matrix $Y$. Is it true that in the worst case, say for some $i$, $\sigma_i \approx 0$, then the convergence speed of $\sigma_i(t)$ will decrease to 0? In that case, can we still call it ‘acceleration’?

- In Proposition 5, the authors build the convergence results for the case $\Lambda_{Q_0} = \lambda_0 I_k$ for the general non-spectral initialization case. However, I hardly can find a non-spectral initialization case satisfying such a condition. Can the authors provide some examples?

---

> ### Author Response · Authors · 2020-11-25
> **Response to AnonReviewer3**
>
> Thank you for the feedback and comments.
>
> - **"Page 2, (2): why call (2) ‘a symmetric one-layer linear model’? Does that suggest $m=n$? The same issue holds for (3). Is $Y$ in (3) the same as that in (2)?"**
>
> Yes, thank you for catching that typo. We have fixed that in the revision.  The matrices are symmetric thus $m=n$ in Section 2.
>
> - **On the difference between convergence rates in the symmetric and asymmetric case and the role of the initialization.**
>
> This is a good observation, thank you. In the (general) asymmetric case, the convergence depends on the conserved quantity $||U_0^T U_0 - V_0^T V_0||$ and also on the data spectrum.  When the initialization is balanced,
> i.e. $\| U_0^TU_0 - V_0^TV_0 \| = 0$, then only the data spectrum survives in the convergence rate. In the symmetric case, one must necessarily have
> balanced initialization (since $V=U$) and this is why the convergence
> rate does not depend on the initialization.
>
> - **What is the convergence behaviour when $\sigma \approx 0$?**
>
> Indeed, it is possible for that scenario to occur when the initial weights are balanced and convergence only depends on the data spectrum. However, as we have shown in the paper, an imbalanced initialization accelerates the convergence of even the smallest components.
>
> - **"In Proposition 5, the authors build the convergence results for the case $\Lambda_{\mathcal{Q}_0} = \lambda_0 I$ for the general non-spectral initialization case. However, I hardly can find a non-spectral initialization case satisfying such a condition. Can the authors provide some examples?"**
>
> We do not expect this condition to be fully satisfied in practice. Such a  condition was a necessary assumption in order to make progress on this problem from
> a mathematical standpoint, namely to obtain closed form solutions that characterizes the dynamics exactly.
> Importantly, however, we have reasons to expect that
> this idealized case does capture a more general behaviour with arbitrary initializations.  This was indeed verified numerically in Section 5 where we illustrate the effect of imbalance. In fact, we did extensive numerical explorations to verify that the dynamics under the assumption $\Lambda_{Q_0} = \lambda_0 I$ are general enough to capture the qualitative behaviour of any asymmetric factorization problem, after the parameter $\lambda_0$ is appropriately tuned (i.e. we claim that the dynamics with an arbitrary initialization can be reproduced by dynamics under
> the assumption $\Lambda_{\mathcal{Q}_0} = \lambda_0 I$ with a suitable
> $\lambda_0$).   In Figure 2 we illustrated this for a few cases.

---

### Official Review · AnonReviewer1 · 2020-10-28
**Interesting Approach to Understanding Implicit Acceleration**

**Rating:** 7
**Confidence:** 5

**Review:**

1.  Paper Summary

This work analyzes the implicit acceleration of gradient flow for over-parameterized 2 layer networks (i.e. 1 hidden layer networks) used for matrix factorization.  By presenting a novel analysis connecting the gradient flow to Riccati type differential equations, this work demonstrates that imbalanced initializations can lead to acceleration.  The authors present convergence rates for symmetric and asymmetric matrix factorization under both spectral and non-spectral initializations.  The convergence rates for these settings are indeed faster than those in the linear model.  The authors lastly provide empirical results to support their theory.


######################################################################

2. Strengths

2.1.  The connection between gradient flow and Riccati type differential equations is novel to the best of my knowledge and provides a simpler and clearer means of understanding implicit acceleration in over-parameterized models than prior works.

2.2. The paper is written clearly and is easy to follow.  The authors present examples of acceleration under the simpler setting of spectral initialization before extending to the more nontrivial case of non-spectral initializations.


######################################################################


3. Limitations

3.1.  I believe the authors are missing some references to related work.  In particular, below are some related works:

(1) https://arxiv.org/pdf/1810.02281.pdf - This work extends the notion of zero balancedness considered in the work by Arora et al. 2018 to the case of approximate balancedness.  The analysis presented in this work is for deep linear networks (more than 2 layers) and for gradient descent.  Technically, I believe this work does fall under the imbalanced case and also yields a fast convergence rate.

(2) https://arxiv.org/abs/2003.06340 - This work analyzes spectral initialization under gradient descent in deep linear networks (of arbitrary layer structure).  In particular, this work also demonstrates linear (fast) convergence for gradient descent under spectral initialization (Proposition 2).

I believe it is important for the authors to position their work with respect to the above works, but I do not feel that missing these references diminishes the novelty of the submitted work.

3.2.  While the connection to Riccati type differential equations is interesting, it would be nice if the authors could discuss how similar analysis could be used for deep linear networks (at the moment there does not appear to be an obvious extension).

3.3.  (Minor) The authors briefly mention that discrete time convergence rates can be derived from continuous time counterparts using symplectic integrators in the introduction, but it would be nice if there were a more rigorous connection between the related work and the rates presented in the submitted work.


######################################################################

4. Score and Rationale

My recommendation is to accept the paper.  I believe the main strength of the paper was in providing a well-presented, rigorous, and novel analysis for understanding acceleration in over-parameterized matrix factorization.  The connection to Riccati type differential equations and identifying dynamical invariants presents an interesting alternative means of gaining intuition around implicit acceleration in over-parameterized networks.


######################################################################

5. Comments

5.1.  I believe the Y in equation 3 should be m x m instead of m x n for the symmetric case.

5.2.  I feel that the notation could be made a bit easier to follow in Section 3 & 4. In particular, I believe bar(U), bar(V) are overloaded to represent diagonal matrices in definition 3, but these quantities are assumed to be non-diagonal for the remainder of the work.  As this is an important point, I feel that it could be emphasized a bit more.

5.3.  I feel that there could be more intermittent references to prior results throughout the work.  For example, the symmetric matrix factorization problem is discussed extensively in https://papers.nips.cc/paper/7195-implicit-regularization-in-matrix-factorization.pdf.  Similarly, invariance and convergence rate for gradient descent and under spectral initialization is discussed in https://arxiv.org/abs/2003.06340.

5.4.  There is an important distinction between the matrix factorization problems considered in this work and prior work.  Namely, the matrix Y has fully observed entries whereas in some prior works, Y is not completely observed.  I think the analysis may get a bit more tricky for the case of unobserved entries, but the authors could maybe point this out.

---

> ### Author Response · Authors · 2020-11-25
> **Response to AnonReviewer1**
>
> We thank the reviewer for the positive feedback and constructive comments which we have taken into account in the revision. Here we address some of the questions raised.
>
> 3.1. Thank you for pointing out these references. We note that our imbalanced setting is more general than the approximate balancedness condition in the first reference. In fact, in the case of gradient flow, convergence is guaranteed and even accelerated as the weights become more imbalanced. The second reference assumes strong alignment which is similar to the spectral initialization assumption and is a particular case of our analysis.
>
> 3.2. The analysis of deeper networks is more involved since weights from different layers start to get "entangled" through the factorization which is manifested through complicated nonlinearities and extra coupled differential equations.
> Specific comments in this direction require a complicated analysis, which can be an interesting (and challenging) problem for future work (as the reviewer already pointed out, it is not obvious how to do this). Nevertheless, the ideas proposed in this paper provide a starting point. Let us mention that, at this stage, we do know that each two consecutive layers will create conserved quantities,  therefore we conjecture that the convergence rate will depend on the accumulated imbalance from each added layer.  Perhaps the basic mechanism  that we introduce in this paper happens in multiple-scale fashion. Anyhow, these comments are still speculative and we do not have a proper mathematical answer.
>
> 3.3. We would like to invite the reviewer to also check the answer that we provided to reviewer 4 regarding a
> related question.
>
> We stress that any  reasonable discretization of a continuous system is expected to capture its behaviour up to some level  of accuracy. Thus, as long as the discretization is stable, one may expect on general terms that the discrete-time convergence rate will be approximately the same as the continuous one (with potentially some source of error which is small).
>  We mentioned this recent work of Franca, Jordan and Vidal (2020) since they propose the first general framework where continuous-time convergence rates
> can be automatically preserved through a class of discretizations (symplectic integrators). This was done for general (dissipative) Hamiltonian systems by exploring the consequences of the underlying symplectic
> structure together with backward-error analysis
> (a powerful technique in numerical analysis of ODEs). Note that these systems are much more
> general than the simple gradient flow considered here, however we mention that gradient flow can be seen
> as a high-friction limit of a classical Hamiltonian system with damping, and
> correspondingly,
> gradient descent can be seen as a high friction limit or a particular
> symplectic integrator.
> In this sense, there is a relation with the mentioned work.
> Nevertheless, the gradient flow is actually much simpler than these general Hamiltonian systems
> and one can show directly that gradient descent closely preserves the gradient flow rates.

---

### Official Review · AnonReviewer4 · 2020-10-29
**Some comparisons may be presented in a more detailed and precise way**

**Rating:** 5
**Confidence:** 4

**Review:**

This paper studies the implicit acceleration of gradient flow for training a two-layer linear model. Compared with the one-layer linear model, the authors show that gradient flow over an overparameterized two-layer linear model may achieve a faster convergence rate, given a nice data spectrum and proper initialization. Moreover, the authors investigate the convergence of gradient flow with an arbitrary initialization and show its connection to Riccati differential equations as well as the explicit regularization. Overall the idea is clearly presented, the experimental results also well back up the theory.

My main concern is that it may not be fair to compare the convergence rate in terms of gradient flow. For example, you can simply reparameterize the parameter by X -> 2X, and the acceleration can also be achieved from the perspective of gradient flow. I think in order to fairly compare the convergence between different parameterizations/initializations, gradient descent is a better choice. Back to the example of X->2X, in this case, the smoothness parameter will become larger, finally one can observe that the convergence rate of GD under this parameterization will remain unchanged.

Regarding the convergence results, the authors still require strong conditions (parameter and data can be diagonalized simultaneously) on the initialization to prove the convergence of gradient flow. What happens if considering more general assumptions on the initialization, such as the random/orthogonal initialization used in the following papers? The authors may also need to compare the convergence rates of the derived results and those in the following papers.

[1] Du, Simon S., and Wei Hu. "Width provably matters in optimization for deep linear neural networks." arXiv preprint arXiv:1901.08572 (2019).

[2] Wu, Lei, Qingcan Wang, and Chao Ma. "Global convergence of gradient descent for deep linear residual networks." Advances in Neural Information Processing Systems. 2019.

[3] Zou, Difan, Philip M. Long, and Quanquan Gu. "On the Global Convergence of Training Deep Linear ResNets". International Conference on Learning Representations.

[4] Hu, Wei, Lechao Xiao, and Jeffrey Pennington. "Provable Benefit of Orthogonal Initialization in Optimizing Deep Linear Networks." International Conference on Learning Representations.

The regularization term in Eq. (20) seems pretty similar to the commonly-used regularization in solving low-rank matrix problems [5,6] for balancing (but they use 1/16 weight), could you please briefly discuss their connections?

[5] Tu, Stephen, et al. "Low-rank solutions of linear matrix equations via Procrustes flow." International Conference on Machine Learning. PMLR, 2016.

[6] Wang, Lingxiao, Xiao Zhang, and Quanquan Gu. "A unified computational and statistical framework for nonconvex low-rank matrix estimation." Artificial Intelligence and Statistics. 2017.

Is the overparameterization necessary? From the presented theorems, I do not see whether the derived results have a dependency on the dimension of the matrix U or V. The authors may clearly specify why one needs to consider overparameterization linear models. For example, assume the data matrix has rank r, can the theory in this paper still hold if set the dimension of U or V as d*r?

=========== after reading rebuttal ===============

I do not agree with the author's response to my first comment. Considering parameterization $X = 2U$ (here I use the notation U to avoid the confusion), then the loss function becomes $L(U) = \frac{1}{2}\\|Y-2U\\| _2^2$ and the gradient flow with respect to $Z$ writes $\dot {U}= -2(2Z-Y)$. Then we have $\dot {X} = 2\dot{U} = -4(2U-Y) = -4(X-Y)$, which gives a rate $O(e^{-4t})$, which is faster than the $O(e^{-t})$ rate achieved without using this parameterization. Therefore, comparing the convergence rate in terms of gradient flow may still not be fair and valid.

Based on this issue I would like to keep my score.

---

> ### Author Response · Authors · 2020-11-25
> **Response to AnonReviewer4**
>
> Thank you for the comments on our paper. Below we address some of the issues raised.
>
> - **On the relevance of continuous-time comparisons of different parameterizations.**
>
> We understand the reviewer's concern but we would like to highlight the fact that a discretization and its underlying ODE are always tied together and one is not free to change one without changing the other. The reviewer's argument of reparametrizing $X \to  2X$ into the ODE is equivalent to also changing the  discrete-time algorithm. The rescaled GF reads $2 \dot{X} = Y - 2X$. Its discretization is $X_{k+1} = X_k + (\eta/2)(Y - 2 X_k)$, which is not the same as the original $X_{k+1} = X_k + \eta (Y - X_k)$. The solution of the rescaled GF is $X(t) = Y/2 + (X(0) - Y/2) e^{-t}$, which gives a rate of $O(e^{-t})$. The discretization yields $X_{k+1} = (1-\eta) X_k + (\eta/2) Y$, which has a convergence rate of $O(e^{- \eta k})$; note that both discrete- and continuous-time rates match.  We see no improvement by this rescaling argument.
>
> To provide more intuition for the reader, we included a specific example in Appendix G in
> the manuscript, showing that the nonlinear GF considered in the paper closely preserves
> the convergence rates of GD under the same problem. We kindly request
> the reviewer to check this discussion.
>
> - **"The convergence results require strong assumptions (parameter and data can be diagonalized simultaneously)... The authors may also need to compare the convergence rates of the derived results and those in the following papers."**
>
> The spectral initialization condition (i.e. where the matrix of parameters and the data can be diagonalized simultaneously) is actually not necessary to prove convergence, and was relaxed in Section 4. Moreover, our results do not make any assumptions on the distribution of the initialization since the only quantity that matters is $||U_0^TU_0 - V_0^TV_0||$. Regarding the listed papers, we have included some of them in the revision (the ones that we find relevant connections). Note that [2] and [3] analyze residual networks and [1] and [4] require sufficiently wide layers to prove convergence. In contrast, our analysis holds as long as the width is larger than the dimensions of the data.
>
> - **On the regularization term.**
>
> Indeed, the regularization term was used in these two references, and the reason was to ensure balancedness and extend the convergence results for symmetric factorizations to asymmetric ones. We noticed this connection as well, and extended it to imbalanced initializations of the type $\Lambda_{\mathcal{Q}_0} = \lambda_0 I$.
> In our work, we rather provide an explanation for the origin of this regularization
> term, namely it arises as a consequence of relaxing the spectral
> initialization condition and requiring that the trajectories of both regularized
> and unregularized problems be the same (this was not a concern in previous works).
> Note that in the general case the dynamics is described by
> Eq. (18), which turns out to be equivalent to a gradient flow applied
> to the objective in Eq. (20).
> In short, while previous works simply introduced this term to enforce
> balancedness, our results give a formal explanation of why the regularization
> has to be in this form (including the $1/8$ factor).
>
> - **Is overparameterization necessary? Can the results be extended to the low-rank case?**
>
> Our results show the effects of overparameterization in the sense of increasing width, i.e. in going from one layer, $X$, to two layers, $UV^T$. There are no constraints on the width, $k$, except that it must be larger
> than $\min(m,n)$ to avoid rank deficiency.
> However, the width $k$ appears indirectly through the quantity $||U_0^TU_0 - V_0^TV_0||$ and it thus may affect the convergence rate. When the data is rank deficient,
> say rank $r$, the dynamics can still be described using Riccati equations and our analysis and results would hold if $k \ge r$, as long as the initial weights are imbalanced; the imbalance offsets the rank deficiency in the Riccati solution.

---

### Official Review · AnonReviewer2 · 2020-11-01
**A detailed analysis of gradient flow in two-layer linear neural networks**

**Rating:** 6
**Confidence:** 4

**Review:**

Summary of review:

This paper provides a detailed analysis of gradient flow in (over-parametrized) two-layer linear neural networks. The main results state the precise dynamics of gradient flow for both symmetric and asymmetric matrix factorization, starting from certain spectral initialization. One novel insight that stems from the analysis is that for asymmetric matrix factorization, "imbalanced initializations", where the left and right singular values of the iterates differ, converges faster than "balanced initializations". Simulations further validate this insight.

Setting:

(i) Symmetric matrix factorization: Given a symmetric matrix Y, the problem is to solve a mean squared loss between Y and UU^T to factorize Y, where U is a (possibly over-parametrized) variable matrix.

(ii) Asymmetric matrix factorization: The asymmetric setting considers an asymmetric matrix Y and the asymmetric factorization of UV^T for factorizing Y.

Results

(i) This paper focuses on the convergence of the gradient flow of U for minimizing the mean squared loss, starting from spectral initializations. Informally, a spectral initialization has the same eigenspace as Y.

For these spectral initializations, the gradient flow paths essentially become coordinate-wise updates over every singular value. Then, the authors went on to state the precise dynamics of gradient flow for every singular value. The results imply that "large" singular values (of Y) converge faster than "small" singular values (of Y) in gradient flow.

(ii) This paper begins by studying spectral initializations, where $||U^TU - V^TV||_F$ is small, then discusses how to generalize their result to non-spectral initializations.

For spectral initializations, this paper observes crucially that $U^TU - V^TV$ is preserved throughout gradient flow. Hence if this quantity starts small, it will remain small throughout gradient flow. Furthermore, for "large" singular values of $U^TU - V^TV$, the results imply that these singular values converge faster than "small" singular values of $U^TU - V^TV$.

For non-spectral initializations, this paper observes that $U^TU - V^TV$ is still preserved during gradient flow, but this quantity now depends on how "balanced" the initializations of U, V are.

Criticism:

It would help improve my understanding of this paper if the authors clarify the following questions.

(a) The "acceleration" claim of this paper comes from comparing the results to a linear model baseline. However, I am not completely sold on this comparison. For example, what would the results imply if compared to properly parametrized U (and V)? Is there any provable advantage of over-parametrization in this setting?

(b) The result of Corollary 1 for symmetric matrix factorization, where larger eigenvalues converge faster than smaller eigenvalues, also appears in linear regression. In particular, the gradient flow of linear regression also shows similar patterns, where larger eigenvalues (of the sample covariance matrix) converge faster than smaller eigenvalues. Therefore, in my opinion, the results in Section 3 and 4 seem more novel compared to the results in Section 2.

(c) While the results state fairly precise dynamics of gradient flow, how well would they extend to settings with sampling errors? For example, what about matrix completion? Would your claim regarding "imbalanced initializations" still hold?

Writing

Overall, this paper is well-written and easy to follow. Although the paper would be easier to read if it is less dense. I do not quite understand the claims in Figure 2. You explained in Section 4 that you need $K \le m + n$ for identity initializations and Proposition 5, but here you say that k ranges from 50 to 200? Please clarify.

---

> ### Author Response · Authors · 2020-11-25
> **Response to AnonReviewer2**
>
> We thank the reviewer for the feedback and comments. Here we address the specific questions that are raised.
>
> **(a) Is there any provable advantage of over-parameterization?**
>
> The goal of the paper is to show the effect of overparameterization by replacing $X$ by its factorized form $UV^T$. As a consequence, the number of parameters increases from $mn$ to $k(m+n)$ with $k \geq \min(m,n)$. Our results show that the convergence rate can depend on the width $k$ of the factorization (the number of columns in $U$ and $V$) indirectly through the quantity $||U^TU - V^TV||$ and the data spectrum; such quantities do not play a role in the non-factorized problem.
> We are not certain about what the reviewer means by 'properly parameterized' $U$ and $V$.
>
> **(b) Sections 3 and 4 are more novel than section 2.**
>
> We agree that the main conclusions of our paper are drawn from Sections 3 and 4. The reason for Section 2 is twofold. First, to provide a warm up and entry point to the reader. Second, to introduce important tools and theorems that will be used in the following sections. For example, in Section 4, we establish a connection between the dynamics of symmetric and asymmetric matrix factorizations and show that the latter can also be reduced to a Riccati equation of the same type as the former. The solution and convergence rate are thus deduced through a similar analysis.
>
> **(c) Can the results extend to settings with sampling error like matrix completion?**
>
> In a matrix completion context we believe the reviewer refers to introducing a linear sampling operator $\mathcal{A}(U V^T)$.
> Note that our formulation has some generality since the conservation law is the result of a symmetry and holds for any objective function of the type $f(UV^T)$. Therefore, we expect that similar conclusions still hold and should extend naturally to the case of introducing such a linear operator.
>
> **"I do not quite understand the claims in Figure 2. You explained in Section 4 that you need $K \leq m+n$ for identity initializations and Proposition 5, but here you say that k ranges from 50 to 200? Please clarify."**
>
> Indeed, in Section 4, we restricted the analysis to the case of a scaled identity imbalance because it is the only setting where we can obtain closed form solutions. However, Figure 2 shows that, even without such an assumption, our results hold empirically to general settings. Moreover, in the bottom row of Figure 2, we show that the condition $\Lambda_{\mathcal{Q}_0} = \lambda_0 I$ is general enough to capture the qualitative behaviour of the dynamics under any type of imbalance. In fact, the figure shows that we can approximate any matrix factorization problem with the dynamics under $\Lambda_{\mathcal{Q}_0} = \lambda_0 I$ for a suitable $\lambda_0$.

---

### Decision · Program_Chairs · 2021-01-07
**Final Decision**

**Decision:**

Reject

**Comment:**

This paper studies the implicit acceleration of gradient flow in over-parameterized two-layer linear models. The authors show that the amount of acceleration depends on the spectrum of the data without assuming small, balanced, or spectral initialization for the weights, and establish interesting connections between matrix factorization and Riccati differential equations.  While this paper provides some interesting results regarding implicit acceleration in training linear neural networks, the reviewers raised quite a few questions and concerns about some claims made in the paper, as well as an inadequate comparison with previous work. Even after the author's response and reviewer discussion,  the reviewers' doubts are still not completely cleared away. I feel the current form of the paper is slightly below the bar of acceptance, and encourage the authors to carefully address reviewers' comments in the revision.